# Exploratory Cortex Metabolic Profiling Revealed the Sedative Effect of Amber in Pentylenetetrazole-Induced Epilepsy-Like Mice

**DOI:** 10.3390/molecules24030460

**Published:** 2019-01-28

**Authors:** Zhenhua Zhu, Chenkai Chen, Yue Zhu, Erxin Shang, Ming Zhao, Sheng Guo, Jianming Guo, Dawei Qian, Zhishu Tang, Hui Yan, Jinao Duan

**Affiliations:** 1Jiangsu Collaborative Innovation Center of Chinese Medicinal Resources Industrialization/State Key Laboratory Cultivation Base for Traditional Chinese Medicine Quality and Efficacy, Nanjing University of Chinese Medicine, Nanjing 210023, China; 04040416@163.com (Z.Z.); maxdone001@126.com (C.C.); zhuyue@njucm.edu.cn (Y.Z.); shex@njutcm.edu.cn (E.S.); mingzhao@njucm.edu.cn (M.Z.); guosheng@njucm.edu.cn (S.G.); njuguo@163.com (J.G.); qiandwnj@126.com (D.Q.); glory-yan@163.com (H.Y.); 2Shaanxi Collaborative Innovation Center of Chinese Medicinal Resources Industrialization, Shaanxi University of Chinese Medicine, Xianyang 712046, China; tzs6565@163.com

**Keywords:** amber, epilepsy, metabolomics, LC/MS, glycerophospholipid metabolism

## Abstract

Epilepsy is a common clinical syndrome characterized by sudden and recurrent attacks and temporary central nervous system dysfunction caused by excessive discharge of neurons in the brain. Amber, a fossilized organic substance formed by the resins of conifers and leguminous plants, was prescribed to tranquilize the mind in China. In this paper, the antiepileptic effect of amber was evaluated by a pentylenetetrazole (PTZ)-induced epileptic model. An untargeted metabolomics approach was applied to investigate metabolic changes in the epileptic model, which was based on HILIC-UHPLC-MS/MS multivariate statistical analysis and metabolism network analysis. The outcome of this study suggested that 35 endogenous metabolites showed marked perturbations. Moreover, four metabolism pathways were mainly involved in epilepsy. After treatment by amber, the endogenous metabolites had a marked tendency to revert back to the situation of the control group which was consistent with phenobarbital. This study characterized the pentylenetetrazole-induced epileptic model and provided new evidence for the sedative effect of amber.

## 1. Introduction

Epilepsy is a common clinical syndrome characterized by sudden and recurrent attacks and temporary central nervous system dysfunction caused by excessive discharge of neurons in the brain. This can result from a variety of endogenous or exogenous factors, often occurring months or years after a sudden injury [1]. A seizure is a form of epilepsy characterized by abnormal movement or behavior caused by unusual electrical activity in the brain. About 70% of epileptic patients who take antiepileptic drugs have complete remission from seizure [2]. The drugs most commonly used for antiepileptic therapy are the benzodiazepines diazepam (oral or rectal), clobazam, buccal or nasal midazolam, lorazepam, phenobarbital, valproic acid, nitrazepam, acetazolamide, chloral hydrate, pyridoxine, and antipyretics [3,4]. Continuous administration of classical antiepileptic drugs has been argued against because of potentially toxic, sedative and cognitive side effects [5,6]. Thus, new antiepileptic medicines are still needed for the remaining one third of epileptic patients that accounted for approximately 1% of the world population [7,8].

Traditional Chinese medicine (TCM) has accumulated ambient experiences in treating epilepsy and mineral Chinese Materia Medica (vermiculitum, chloriti lapis, amber, etc.) has a long history of use for mind-tranquilizing with significant effects [9]. Amber is widely used in the treatment of epilepsy in traditional Chinese medicine [10,11,12]. Amber is a fossilized organic substance formed by the resins of conifers and leguminous plants which grew during the Mesozoic cretaceous to Cenozoic tertiary through complex geological processes [13,14]. Amber is also admixed with monoterpenoids, sesquiterpenoids, diterpenoids, triterpenoids, leaf wax and biopolymer products indicating mixed sources of medicinal materials [15,16]. TCM doctors observe that amber possesses the character of tranquilizing the mind, promoting the blood circulation to remove blood stasis, promoting diuresis and relieving strangury. In clinics, amber is used for treating psychological diseases with the symptoms of restfulness, convulsion and epilepsy. It was first recorded in Shen Nong’s Herbal Classic (before A.D. 25) and has been used in TCM clinics for thousands of years. In modern studies, kujiol A and kujigamberol B isolated from Kuji Amber were proven to be Ca^2+^-signal transduction inhibitors [17,18]. When Ca^2+^-signal transduction is involved in an allergy, 3-cler-oden-15-oic acid from Dominican amber can inhibit Ca^2+^-influx [19,20]. However, to date, no modern pharmacological studies on the anticonvulsive effect of amber have been reported and the action mechanism of amber in treating epilepsy has never been elucidated.

Metabolomics is a top–down systems biology approach which is the systematic study of the special chemical fingerprints left by specific cellular processes and the overall study of their small-molecule metabolites [21]. Analytical technologies, such as ^1^H-NMR spectroscopy [22,23] and mass spectrometry (MS) [24,25,26] (the latter mainly combined with separation techniques, for example, liquid chromatography (LC) [27,28,29], gas chromatography (GC) [30,31], or capillary electrophoresis (CE) [32] are typically used for untargeted metabolic studies. 

In this experiment, the PTZ-induced epilepsy mice were treated with amber, and behavioral observation showed that there was a significant trend toward the normal after the intervention. The cerebral cortex of ICR mice was analyzed and determined by UHPLC-MS/MS. The potential biomarkers and metabolic pathways of amber anti-epileptic treatment were searched using a mode discrimination method and metabolome network database. Metabolomics was used to explain the metabolic pathway and mechanism of amber in the treatment of epilepsy in mice.

## 2. Results and Discussion

### 2.1. Behavioral Analysis

Behavioral scoring is commonly used in different seizure models to assess seizure intensity. Racine’s scale is frequently used as an intensity measurement in other experimental seizure or epilepsy models which originally developed for the amygdala-kindling model [33,34].

The mice in the model group showed obvious epileptic symptoms after receiving an intraperitoneal injection of pentylenetetrazole (PTZ), and the Racine score of epilepsy reached 4 or 5. As shown in Figure 1, the positive drug phenobarbital (PB) could reverse the epileptic symptoms, such as incubation time and seizures, indicating that the established animal model was suitable for anti-epileptic activity screening.

Under the intervention of amber, the incubation period was significantly prolonged, and the level of seizures was reduced as well (*p* < 0.01, Figure 1A,B).

### 2.2. Brain Tissue Nissl Staining

The Nissl staining, a classic nucleic acid staining method to observe the damage degree in the cortex and hippocampal neurons of mice, was widely used in the study of epilepsy [35,36]. To evaluate the effect of amber on cell death in the PTZ-induced kindling model, Nissl staining was performed. The neuronal cells of the control group were found to be round or conical while the model group was impaired. The average density of intact surviving neurons was lower in the PTZ group compared to the control group, while pre-treatment with amber reversed the damage to cell morphology (Figure 2).

### 2.3. Analysis of Metabolite Profiling

The cortex samples were analyzed by ESI-MS under positive and negative ion modes. Base peak chromatograms (BPCs) for different groups are shown in Figure 3. In order to reveal the differences, peak extraction, peak alignment, background deduction, and elimination of missing values by the 80% rule of zero were carried out for the data of each group. After data filtering, 408 metabolites from positive and negative ion modes were used to build multivariate models separately.

### 2.4. Multivariate Data Analysis

Representative HILIC-UHPLC-ORBITRAP-MS cortex metabolic profiles from a control and a PTZ-treated animal, in ESI^+^ and ESI^−^ modes, are shown in Figure 3. The use of quality control (QC) samples and evaluation of data quality have been detailed previously for metabolomics analyses of biological samples [21]. Clustering of QC samples was assessed using principal component analysis (PCA) to reveal if platform stability had been achieved. A PCA scores plot (PC1 vs. PC2) of all study cortex and QC samples analyzed in ESI^+^ mode and ESI^−^ mode are shown in Figure 4A (ESI^+^) and 4B (ESI^−^). The QC samples are clustered, indicating good reproducibility of the data. 

PCA analysis in positive and negative TIC was used to evaluate the PTZ-induced epileptic model. In the positive ion mode, a model with two principal components was obtained (R2X cum = 0.641, Q2 cum = 0.461) while three principal components were obtained in negative mode (R2X cum = 0.836, Q2 cum = 0.449). PCA scores showed that all samples in the positive ion mode were distributed in the 95% confidence interval ellipse while one sample was distributed out of the ellipse in negative mode. 

As shown in the PCA scores scatter plot, significant separation between the three groups in the unsupervised mode could be observed, indicating the difference in the metabolites between the different groups.

### 2.5. Comparison of Model Against Control Using OPLS-DA

When the amber group was omitted, six model samples compared against five control samples produced a strong Orthogonal Projections to Latent Structures Discriminant Analysis(OPLS-DA) model (Figure 5A,B, R2X cum = 0.976, Q2 cum = 0.867) explaining 97.6% of the variation in the samples with seven components in positive mode and 74.6% of the variation with two components in negative mode (Figure 5C,D, R2X cum = 0.746, Q2 cum = 0.895). Q2 > 0.5 is generally accepted as being indicative of a robust model and the model gave a permutations plot where all the permutated Q2 values (*n* = 200) on the left are lower than the points on the right (Figure 5B,D) and the line plot intercepts the y-axis below 0 [34,35].

### 2.6. Screening and Identification of Metabolic Differences

In this study, loading S-plots (Figure 6) and the variable importance (VIP) generated by OPLS–DA analysis in the projection were used to select potential biomarkers. VIP values larger than 1 were considered to be more important on the classification than average. Ions with *p* < 0.05 (using an independent sample t-test) showing significant changes in the model group compared to the control group were taken as candidate biomarkers [37]. For those differential features, theoretical database searching and manual spectrum confirmation were used for identification. Thirty-five metabolites differentially expressed between the control and model groups (Table 1) were identified.

### 2.7. Metabolic Pathway Analysis

Pathway analysis of the discriminating metabolites was performed with MetaboAnalyst 4.0, a web-based tool for pathway analysis and visualization of metabolomics [38]. As shown in Figure 7, biological pathway analysis revealed that the identified metabolites important for epilepsy are mainly responsible for the following metabolism pathways: (A) glycerophospholipid metabolism; (B) nicotinate and nicotinamide metabolism; (C) alanine, aspartate and glutamate metabolism; and (D) pyruvate metabolism. The trends of the metabolites associated with the above four pathways are shown in Figure 8.

The variation of phosphorylcholine, choline, acetylcholine, phosphatidylcholine (PC) and lysophosphatidylcholine (LysoPC) in the cortex could potentially indicate that the glycerophospholipid metabolism was disrupted and played a major role in seizures. Increased choline may reflect myelin breakdown, increased cell density, or gliosis, which may indicate Alzheimer’s disease or epilepsy. Phosphatidylcholine is the main component of the cell membrane and usually exists on the surface of the ectoplasmic membrane. Excitotoxic events enhance the hydrolysis of phosphatidylcholine in the brain, which was evidenced caused by a concomitant increase in the levels of choline and free fatty acids [39].

The variation of gamma-aminobutyric acid (GABA) in the cortex could potentially reflect that the metabolism of alanine, aspartate and glutamate is disrupted, which plays a major role in seizures. GABA, a key inhibitory neurotransmitter, is synthesized through the decarboxylation of glutamate via alanine, aspartate and glutamate metabolism [1]. Studies have shown that epileptic-related brain damage is caused by the release of excitatory amino acid neurotransmitters from over-discharged presynaptic terminals that eventually reach neurotoxic concentrations [40]. Evidence indicated that there are certain regions of the brain where enhanced GABA transmission is anticonvulsant [41]. The GABA level in the cortex was lower in the epileptic group than those in the healthy group which supported the literature [1,42]. After the amber intervention, the level of GABA rebounded and approached the control group.

## 3. Materials and Methods 

### 3.1. Materials and Extract Preparation

Acetonitrile, methanol, alcohol, acetone and formic acid (HPLC grade) were purchased from Merck Company Inc. (Darmstadt, Germany); HPLC grade water was produced by a Direct-Q3 Ultrapure Water System from Millipore (Hertfordshire, UK). Other reagents and chemicals were of analytical grade.

Amber samples were purchased from Shaanxi Science Pharmaceutical Co. Ltd. They were identified by Dr. Hui Yan (Department of Medicinal Plants, Nanjing University of Chinese Medicine, Nanjing, China). Pentylenetetrazol (PTZ), Phenobarbital (PB), and sodium chloride injections were from Macklin (Shanghai, China), Shanxi Yunpeng Pharmaceutical Co. Ltd. (Linfen, China), and Chenxin Pharmaceutical Co. Ltd (Qidong, China), respectively.

### 3.2. In Vivo Experiments Protocol

Specific Pathogen-Free (SPF) male ICR mice were purchased from the Experimental Animal Center of Qinglongshan (Nanjing, China, license number: 2018-0001). All the mice were kept in the Specific Pathogen Free Center of Nanjing University of Chinese Medicine, Nanjing, China. All animal studies were in accordance with the guidelines of the Animal Ethics Committee of Nanjing University of Chinese Medicine (201810A016).

After an initial acclimation period of 7 days in cages, 40 mice were randomly allocated into 4 groups: Control, model (oral administration of water for 14 days followed by intraperitoneal injection of PTZ at the dose of 60 mg/kg), PB (PTZ + phenobarbital, abdominal injection with phenobarbital at a dose of 40 mg/kg followed by an intraperitoneal injection of PTZ at the dose of 60 mg/kg with 30 min interval), and amber (oral administration of amber for 14 days at the dose of 0.9 g/kg followed by an intraperitoneal injection of PTZ at the dose of 60 mg/kg).

After the intraperitoneal injection of pentylenetetrazole, observation of the seizures lasted for 30 min, and the severity, latency and duration of seizures were recorded. The degree of epileptic behavior was divided into 6 grades according to the Racine standard of neurology: Level 0, no response; Level I, ear and facial twitch; Level II, myoclonus, but no upright position; Level III, myoclonus, with axial position; Level IV, systemic tonic-clonic seizure; and Level V, systemic tonic-clonic seizure and loss of postural control [43].

The mice were sacrificed after the observation. Three brains of each group were taken and immersed in 4% paraformaldehyde, and paraffin sections of coronal plane were used for Nissl staining. The rest of the brain was divided into cortex and hippocampus, and kept in liquid nitrogen. 

After thawing in the fridge at 4 °C, 40 mg of the cortex sample was precisely weighed and used for the following process: 160 μL of extract solution, vortexed for 1 min, sonicated for 5 min in an ice bath, centrifuged for 10 min under 13,000 rpm at 4 °C, and then 100 μL of supernatant was collected and concentrated to dry. The extract was then reconstituted with 40 μL of mobile phase, vortexed for 1 min, sonicated for 5 min in ice bath and centrifuged for 10 min under 13,000 rpm at 4 °C. The supernatant was then finally collected for analysis. Quality control (QC) samples were prepared by pooling aliquots (2 μL) of each sample.

### 3.3. UHPLC–LTQ–Orbitrap MS Analyses

An LTQ–Orbitrap Velos pro-mass spectrometer (Thermo Scientific) equipped with an ESI source was set to collect data from *m/z* 100 to 1000 in profile mode. External calibration was carried out with a standard LTQ calibration mixture (Thermo Scientific, Waltham, MA, USA). The following settings were used for MS detection: vaporizer temperature, 280 °C; sheath and auxiliary gases, 35 and 15 (arbitrary units); spray voltage, 3.5 kV(ESI^+^), 2.5 kV(ESI^−^); capillary temperature, 350 °C; capillary voltage, 10 V; tube-lens voltage, 120 V; maximum injection time, 1000 ms; maximum number of ions collected for each scan, 5 × 10^5^; mass resolution, 30,000; MS/MS mode, high-energy induced dissociation (HCD); Collision gas, N_2_.

For LC separation, UHPLC Dionex Ultimate 3000 (Thermo Scientific, San Jose, CA, USA) and an ACQUITYTM UPLC BEH Amide column (1.7 μm, 2.1 mm × 100 mm) were used. Water and acetonitrile modified with 5 mM ammonium formate, 5 mM ammonium acetate, and 0.1% formic acid were used as mobile phase A and B, respectively. The column was eluted with a program as follows: The percentage of B was decreased from 95% to 55% at the first 13 min, and then held for 2 min. The flow rate and injection volume were set at 0.4 mL/min and 2 μL, respectively.

### 3.4. Data Analysis

The experimental data were analyzed by R and Compound Discoverer 2.1 software (Thermo Fisher Scientific, Waltham, MA, USA), including peak extraction, peak alignment, background deduction and compound identification. After removing exogenous component interference, the extracted ion fragment peak area was normalized by SIMCA 14.1 software (Umetrics AB, Umea, Sweden), and multivariate statistical analysis was conducted after standardization. The mean-centering method and pareto-scaling method were used to transform the data, and the importance of low-abundance ions was increased, while the noise was not obviously amplified. Principal component analysis (PCA) and orthogonal partial least squares discriminant analysis (OPLS-DA) were used for classification. The s-plot was generated to discover the significant components between groups as potential markers.

The potential endogenous biomarkers were identified based on accurate molecular mass, MS/MS fragments, and retention behavior by searching online databases. The potential markers were identified within 5 ppm. Moreover, the MS/MS spectrum match was searched in the METLIN database. In this study, the Compound Discoverer (Thermo Fisher Scientific, Inc., Waltham, MA, USA) was used to search KEGG, HMDB and LIPID MAPS database. Metabolism pathway analysis was performed with MetaboAnalyst 4.0 (McGill University, Montreal, QC, Canada), a web-based tool for pathway analysis and visualization metabolomics.

## 4. Conclusions

Metabolomics, as a systematic method, could systematically identify, quantify or reveal the metabolites of diseases, provide a basis for the diagnosis, biomarkers and/or monitoring tools of diseases, and provide potential targets for the treatment and prevention of diseases. In this study, the metabolic changes and potential biomarkers of epileptic models were studied by using the metabolomics method of LC-MS technology and metabolic network analysis. After intervention of amber, the incubation period was significantly prolonged, and the level of seizures was reduced as well. The damage to the cortex and hippocampal neuron cells was reversed, the fluctuating composition metabolites had a marked tendency to revert back to the control group which was consistent with phenobarbital. This study characterized the PTZ-induced epileptic model and provided new evidences for amber sedative effect. Our work enhanced not only the understanding of the pathology of epilepsy, but also revealed that amber could effectively inhibit seizures with a similar mechanism to that of phenobarbital.

## Figures and Tables

**Figure 1 molecules-24-00460-f001:**
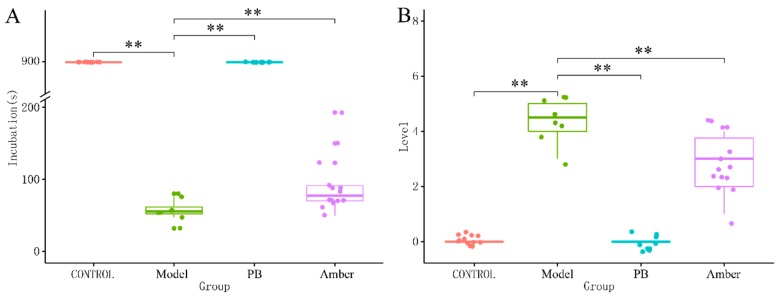
The effect of amber on behavior in the seizure model induced by pentylenetetrazole. (**A**) The incubation time of each group. Control group and PB group had no incubation time, and used 900 s for analysis. Under the intervention of amber, the incubation period was significantly prolonged compared to the model group. (**B**) Under the intervention of amber, the level of seizures was significantly reduced compared to the model group. ** *p* < 0.01 for extremely significant difference.

**Figure 2 molecules-24-00460-f002:**
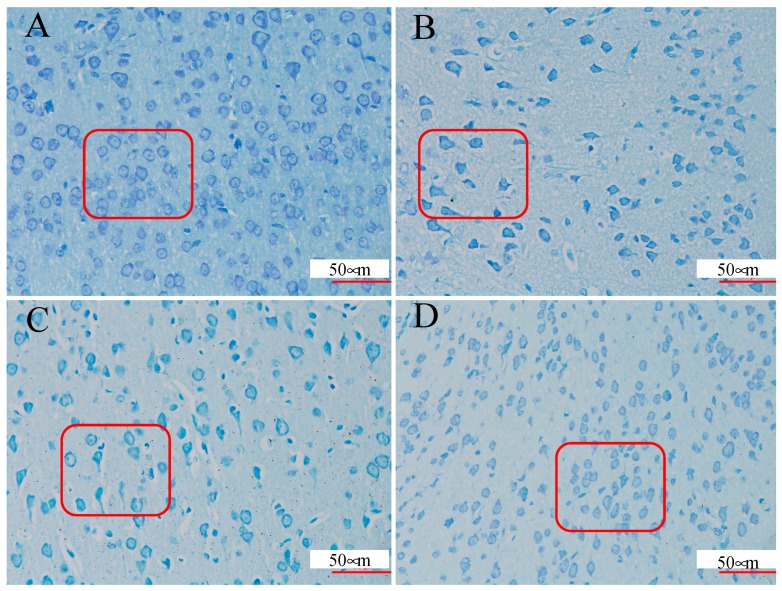
Amber rescues CA1 pyramidal neurons from seizure-induced damage as revealed by Nissl staining. (**A**) Control group; (**B**) Model group (PTZ); (**C**) PB group (PTZ + Phenobarbital); (**D**) Amber group (PTZ + amber). Photomicrographs show sample CA1 subfields (magnification, ×400) in the coronal plane for each treatment group. A damaged cell body is indicated by red frame. These signs of neural damage were reduced by amber pre-treatment.

**Figure 3 molecules-24-00460-f003:**
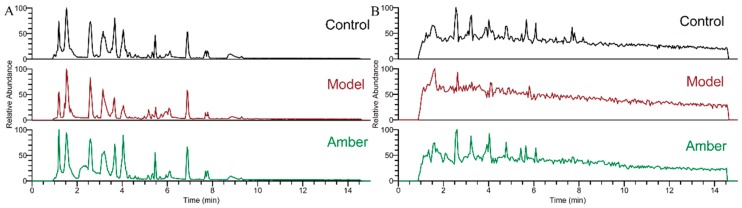
Base peak chromatograms (BPCs) of cortex samples of control group, model group, and amber group under ESI^+^ (**A**) and ESI^−^ (**B**) MS conditions.

**Figure 4 molecules-24-00460-f004:**
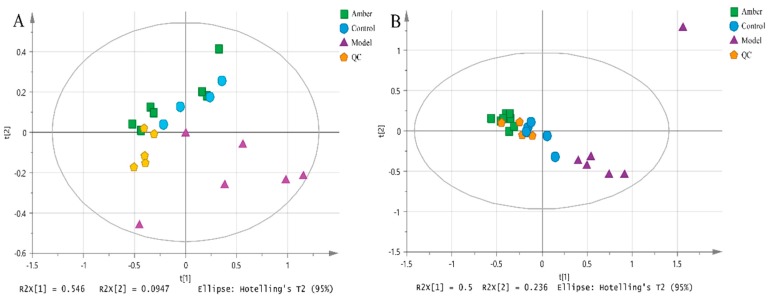
PCA scores scatter plot for control group, model group and amber group. (**A**) positive mode, R2X (cum) = 0.641, Q2 (cum) = 0.461; (**B**) negative mode, R2X (cum) = 0.836, Q2 (cum) = 0.449.

**Figure 5 molecules-24-00460-f005:**
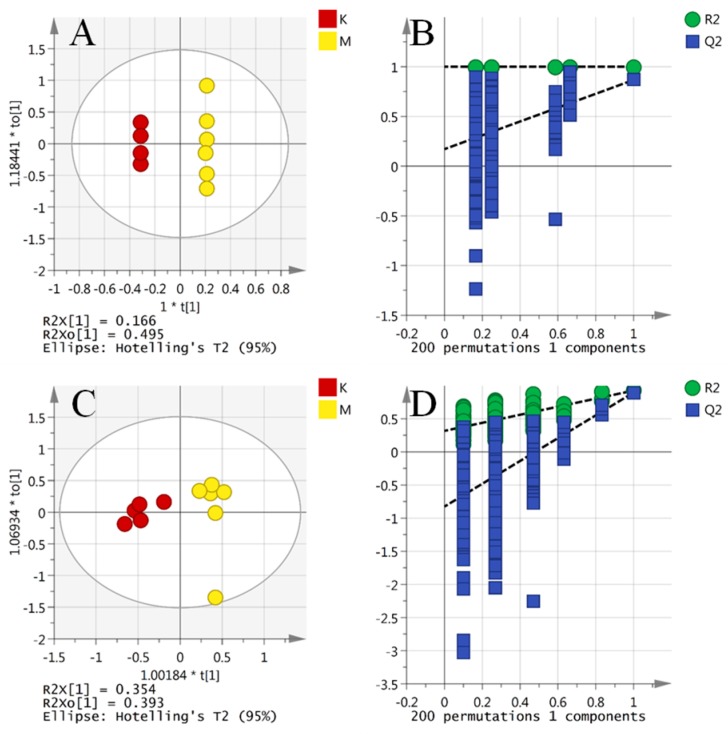
OPLS-DA score plot and validation plot of the OPLS-DA model of control group and model group. OPLS-DA score plot for the first two components showed the separation between the control group and model group. The fitness (R2Y) and prediction power (Q2Y) of this two-component model were 0.979 and 0.918, respectively ((**A**) ESI^+^; (**C**) ESI^−^). Validation plot of the OPLS-DA model of control group and model group were obtained from 200 permutation tests. The intercepts of R2 were lower than the original point to the right, whereas those of Q2 were negative, indicating no signs of overfit ((**B**) ESI^+^; (**D**) ESI^−^).

**Figure 6 molecules-24-00460-f006:**
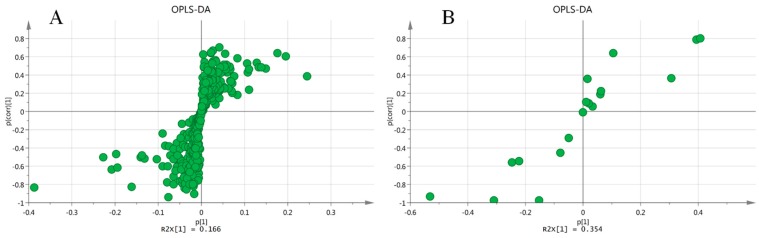
Loading S-plots generated by OPLS-DA analysis in positive mode (**A**) and negative mode (**B**). The x-axis is a measure of the relative abundance of ions, and the y-axis is a measure of the correlation of each ion to the model.

**Figure 7 molecules-24-00460-f007:**
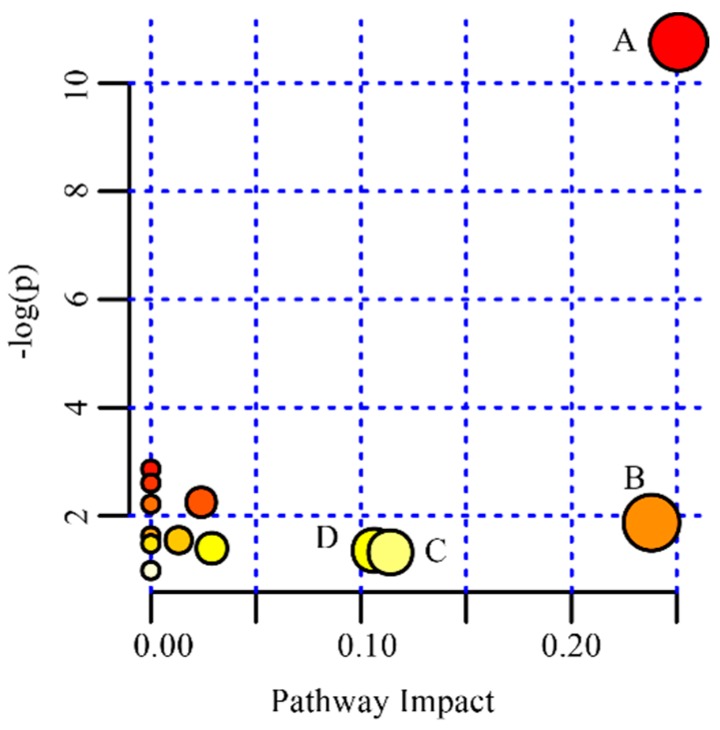
Summary of pathway analysis with MetPA. (**A**) glycerophospholipid metabolism; (**B**) nicotinate and nicotinamide metabolism; (**C**) alanine, aspartate and glutamate metabolism; (**D**) pyruvate metabolism.

**Figure 8 molecules-24-00460-f008:**
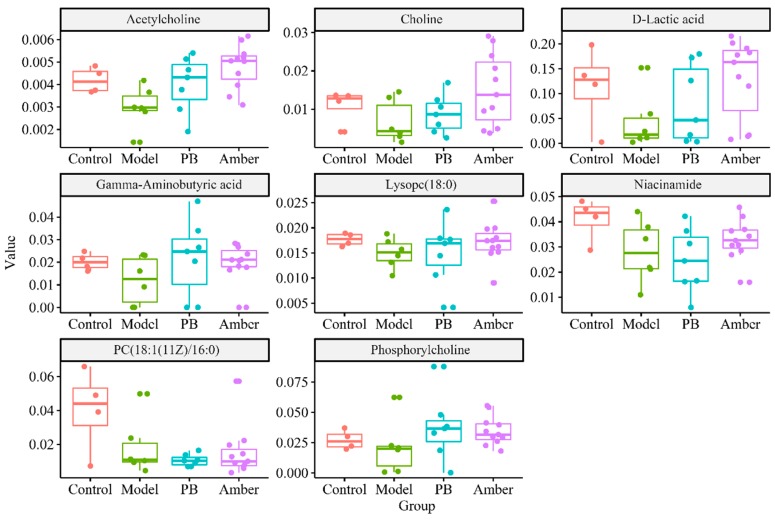
Boxplot of the metabolites associated with the four pathways in Figure 7.

**Table 1 molecules-24-00460-t001:** Cortical differential metabolites between the control group and the model group.

No.	HMDB ID	Metabolites	Formula	*m/z*	ESI Mode	*p* Value	Ratio Control/Model
**1**	HMDB0010404	Lysopc(22:6(4z,7z,10z,13z,16z,19z))	C_30_H_50_NO_7_P	568.3398	+	<0.001	4.29
**2**	HMDB0001406	Niacinamide	C_6_H_6_N_2_O	123.0553	+	<0.001	2.64
**3**	HMDB0002815	Lysopc(18:1(9z))	C_26_H_52_NO_7_P	522.3554	+	<0.001	2.95
**4**	HMDB0000895	Acetylcholine	C_7_H_16_NO_2_	146.1181	+	<0.001	2.49
**5**	HMDB0000157	Hypoxanthine	C_5_H_4_N_4_O	137.0458	+	<0.001	2.56
**6**	HMDB0062176	*N*-lactoyl-leucine	C_9_H_17_NO_4_	204.1230	+	<0.001	2.52
**7**	HMDB0000201	Acetylcarnitine	C_9_H_17_NO_4_	204.1230	+	<0.001	2.26
**8**	HMDB0010396	Lysopc(20:4(8z,11z,14z,17z))	C_28_H_50_NO_7_P	544.3398	+	<0.001	3.03
**9**	HMDB0144632	diethylamino-Acetaldehyde	C_6_H_13_NO	116.1070	+	<0.001	2.47
**10**	HMDB0010382	Lysopc(16:0)	C_24_H_50_NO_7_P	496.3398	+	<0.001	3.32
**11**	HMDB0031779	Isoprothiolane	C_12_H_18_O_4_S_2_	291.0719	+	<0.001	2.28
**12**	HMDB0000195	Inosine	C_10_H_12_N_4_O_5_	269.0880	+	<0.002	2.33
**13**	HMDB0000562	Creatinine	C_4_H_7_N_3_O	114.0662	+	<0.003	2.51
**14**	HMDB0011496	Lysope(0:0/22:6(4z,7z,10z,13z,16z,19z))	C_27_H_44_NO_7_P	526.2928	+	<0.004	3.12
**15**	HMDB0000062	l-carnitine	C_7_H_15_NO_3_	162.1125	+	0.0015	3.17
**16**	HMDB0010720	butenoic acid	C_4_H_6_O_2_	87.04406	+	0.0017	3.06
**17**	HMDB0000112	Gamma-aminobutyric acid(GABA)	C_4_H_9_NO_2_	104.0706	+	0.0018	3.18
**18**	HMDB0010384	Lysopc(18:0)	C_26_H_54_NO_7_P	524.3711	+	0.0018	3.05
**19**	HMDB0038039	Isovaleric acid amine	C_5_H_13_NO_2_	120.1019	+	0.0025	2.16
**20**	HMDB0060348	2-Maleylacetate	C_6_H_6_O_5_	159.0288	+	0.0028	2.58
**21**	HMDB0000064	Creatine	C_4_H_9_N_3_O_2_	132.0768	+	0.0029	2.31
**22**	HMDB0005065	Oleoyl carnitine	C_25_H_47_NO_4_	426.3578	+	0.0041	1.59
**23**	HMDB0001565	Phosphorylcholine	C_5_H_15_NO_4_P	184.0739	+	0.0105	2.96
**24**	HMDB0010382	Lysopc(16:0)	C_24_H_50_NO_7_P	496.3398	+	0.0110	8.87
**25**	HMDB0000097	Choline	C_5_H_14_NO	104.1075	+	0.0211	2.45
**26**	HMDB0008067	Pc(18:1(11z)/16:0)	C_42_H_82_NO_8_P	760.5851	+	0.0228	4.07
**27**	HMDB0008003	Pc(16:1(9z)/18:0)	C_42_H_82_NO_8_P	760.5851	+	0.0329	2.56
**28**	HMDB0007911	Pc(14:1(9z)/20:0)	C_42_H_82_NO_8_P	760.5851	+	0.0405	4.29
**29**	HMDB0007879	Pc(14:0/20:1(11z))	C_42_H_82_NO_8_P	760.5851	+	0.0465	3.05
**30**	HMDB0001406	Niacinamide	C_6_H_6_N_2_O	123.0553	+	<0.001	4.29
**31**	HMDB0001311	d-lactic acid	C_3_H_6_O_3_	89.0244	−	0.0981	3.86
**32**	HMDB0000805	Pyrrolidonecarboxylic acid	C_5_H_7_NO_3_	128.0353	−	0.0529	2.43
**33**	HMDB0035291	Isoplumbagin	C_11_H_8_O_3_	187.0401	−	0.0103	3.62
**34**	HMDB0014118	Trifluoroacetic acid	C_2_HF_3_O_2_	112.9856	−	0.0327	1.82
**35**	HMDB0000700	Hydroxypropionic acid	C_3_H_6_O_3_	89.0244	−	0.0368	0.62

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
