# Peer review of "Exploratory Cortex Metabolic Profiling Revealed the Sedative Effect of Amber in Pentylenetetrazole-Induced Epilepsy-Like Mice"

_molecules, 2019, doi:10.3390/molecules24030460_

Round 1
Reviewer 1 Report
The manuscript Exploratory Cortex Metabolic Profiling Revealed the Sedative Effect of Amber in pentylenetetrazole-induced Epilepsy-like Mice
fit the journal's aims and scopes.
The topic is actual and the experimental design is sufficiently clear.
The main weakness are the complete non information on the amber content. Authors need to add some charachterizazion of the product. At least an LC-MS and/or GC-MS or other phytochemical profiling approach is absolutely needed!
The doses: how authors decided doses? 9 g/Kg on the basis of what?
The authors did not consdider any other tissue for their analysis?
some furthere discussion can be added also comparing with other literature
The introduction need a careful revision and change in order to assess more details in amber treatments, as well as in other TCM treatment for epilepsy. Reference and introduction and literature verification for introduction are not sufficient and need revision and reformulation.
Aim of the work need also to be clearly rewritten
English form expecially in the introduction and discussion need a very careful revision
I also enclose pdf with annotation
the manuscript need to be implemented

Author Response
Dear reviewer:
Thank you very much for your time and suggestions.
We have carefully revised the manuscript according to all the comments and suggestions. And the detailed responses to the comments was attached below for your reference.
Looking forward to hearing from you soon, and
Wishing a healthy, happy, prosperous, and blessed New Year of 2019!
Jin-ao Duan

Reviewer 2 Report
This is descriptive study coming under the category of metbonomics- a subcategory of metabolomics of Amber which is used in Traditional Chinese Medicine.
In this paper, the antiepileptic effect of amber was evaluated by pentylenetetrazole(PTZ)-induced epileptic model.
Minor revision
1. Page 1, line 36 . The drugs most commonly used for antiepileptic THERAPY instead of drugs
2. Page 2, line 49. relieving straguria- may be changed to STRANGURY
3. Page 3, line 91. Please clarify this sentence- The neuronal cells of the control group were found to be round or conical while the model group was incomplete. What is meant by incomplete?
4. Page 3, line 93. The average density of intact surviving neurons was lower in the PTZ group
compared to the control group, while pre-treatment with amber reversed the damage to cell
morphology (Figure 2). Please QUANTIFY the neuronal damage.
5. Page Table 1. Cortical differential metabolites between the control group and the model group.
It is better to show the percentage of decrease between the control group and the treatment group and the degree of statistical significance.
Author Response

(The authors gave the same response as above.)

Round 2
Reviewer 1 Report
The authors have improved the manuscript and all my previous comments were satisfactory resolved thus in the present form the paper in my opinion is suitable for publication